# Hierarchical Molecular Representation Learning via Fragment-Based Self-Supervised Embedding Prediction

## Abstract

Graph self-supervised learning (GSSL) has demonstrated strong potential for generating expressive graph embeddings without the need for human annotations, making it particularly valuable in domains with high labeling costs such as molecular graph analysis. However, existing GSSL methods mostly focus on node- or edge-level information, often ignoring chemically relevant substructures which strongly influence molecular properties. In this work, we propose **Gra**ph **S**emantic **P**redictive **Net**work (GraSPNet), a hierarchical architecture that predicts both node and semantically meaningful fragments of a graph in the embedding space. GraSPNet decomposes molecular graphs into meaningful fragments without relying on predefined chemical vocabulary and learns graph representations through message-passing graph neural networks. It further captures fragment-level semantics by encoding fragment information and modeling interactions through node-fragment and fragment-fragment message passing. By performing masked prediction of node and fragment features in semantic space, GraSPNet captures structural information at multiple resolutions. Experiments show that GraSPNet is both expressive and generalizable, outperforming existing state-of-the-art methods on multiple molecular property prediction benchmarks in transfer learning settings. The code will be released upon acceptance.

## 1 Introduction

Graphs are a powerful tool for representing structured and complex non-Euclidean data in the real world, as they naturally capture relationships and dependencies between entities (Ma & Tang, 2021; Bacciu et al., 2020). Techniques such as Graph Neural Networks (GNNs) have demonstrated notable success in this context, enabling models to capture both local and global structural information (Corso et al., 2024). Molecules are inherently graph-structured data, where atoms and chemical bonds are represented as nodes and edges, respectively. Molecular representation learning focuses on deriving meaningful embeddings of molecular graphs, forming the foundation for a wide range of applications, including molecular property prediction (You et al., 2020), drug discovery (Gilmer et al., 2017), and retrosynthesis (Yan et al., 2020). However, the costly process of labeling molecular properties and the scarcity of task-specific annotations underscore the need for self-supervised learning approaches in molecular representation.

In self-supervised learning, supervisory signals are derived directly from the input data, typically following either invariance-based or generative-based paradigms (Liu et al., 2021). These approaches aim to learn mutual information between different views of a graph by applying node-level and edge-level augmentations (Zhang et al., 2021a), or to reconstruct certain characteristics of the graph from original or corrupted inputs (Kipf & Welling, 2016b; Hou et al., 2022). However, for graph classification tasks such as molecular property prediction, chemically meaningful substructures—such as functional groups—often play a decisive role in determining molecular behavior (Ju et al., 2023). While reconstructing atoms (nodes) and bonds (edges) helps capture local structure, it may fail to encode higher-level semantics that are critical for downstream tasks requiring a deeper understanding of molecular graphs.

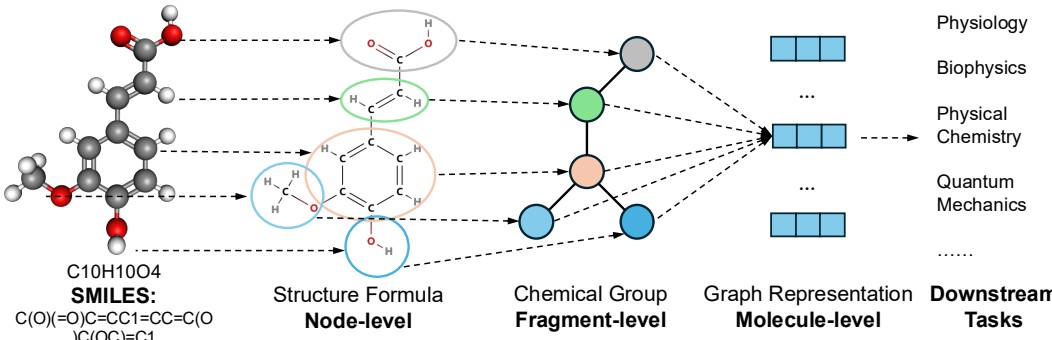

Figure 1: An example of hierarchical representation learning on molecular graph. The molecule is represented as a string-based notations (SMILES) and encoded at three semantic levels—node (atoms), fragment (e.g., functional groups), and graph to support various downstream tasks.

The main challenge lies in preserving different levels of semantic information within graph data. Figure 1 illustrates the hierarchy of representation learning in molecular graph analysis, progressing from chemical representations to graph-level embeddings and ultimately to downstream tasks. In other domains, such as natural language processing and computer vision, tokenizer is commonly used to divide input data into smaller units—such as words or pixel patches—that encapsulate semantically meaningful information for representation learning (Vaswani et al., 2017; Dosovitskiy et al., 2021). Methods that adaptively adjust the size and position of patches to capture richer semantic regions from images further underscore the importance of semantic preservation in data (Chen et al., 2021). In graphs composed of nodes with relational connections, clusters of interconnected nodes often carry significant information and serve as key indicators of the graph's overall characteristics (Milo et al., 2002). Therefore, the exploration of fragment-level semantic information in graphs is of great importance. Although fragment-based representation learning techniques have been proposed, they often rely on discrete fragment counting (Kashtan et al., 2004), which may lack chemical validity, or on generative processes that are computationally intensive (Tang et al., 2020; Zhang et al., 2021b). Consequently, a more efficient and generalized method for capturing rich semantic information in graphs remains largely unexplored.

In this work, we focus on the problem of extracting semantically rich representations from large pre-trained molecular structure datasets. We aim to address the following question: How can we better utilize the hierarchical semantic information in large unlabeled molecule graph databases and generalize to various downstream tasks? We propose **Gra**ph **S**emantic **P**redictive **Net**work (**GraSPNet**), a hierarchical pretraining method based on predicting the representations of nodes and fragments without relying on pre-defined chemical vocabularies. In contrast to previous approaches that focus on reconstructing node-level inputs or predicting node representations, GraSPNet utilizes both nodes and fragments representation as prediction targets, benefiting from information at multiple resolutions. By incorporating a hierarchical context and target encoder, the model is guided to learn representations that capture multiple levels of semantic information, including fine-grained node features and structures, node–fragment dependencies, and coarse-grained fragment patterns.

## 2 RELATED WORK

**Pretraining on Graphs.** GNNs are a class of deep learning models designed to capture complex relationships by aggregating the features of the local neighbors of the node through neural networks (Hamilton et al., 2017; Kipf & Welling, 2016a). To alleviate the generalization problem of graph-based learning, graph pretraining has been actively explored to benefit from large databases of unlabeled graph data. Pretraining methods on graphs can be categorized as contrastive methods and generative methods. Graph contrastive learning methods (You et al., 2020; Zhu et al., 2021; You et al., 2021; Zhao et al., 2021; Yu et al., 2022) learn invariant representations under graph augmentations, while generative methods like Graph Autoencoders (Hinton & Zemel, 1993; Pan et al., 2018; Wang et al., 2017; Park et al., 2019; Salehi & Davulcu, 2020) rely on reconstruction objec-

tives from the input graph. Recently, masked autoencoder frameworks (He et al., 2022) including GraphMAE (Hou et al., 2022), S2GAE (Tan et al., 2023), MaskGAE (Li et al., 2023) where certain node or edge attributes are perturbed and encoders and decoders are trained to reconstruct them with the remaining graph.

**Fragment-based GNN.** Representation learning on molecules has made use of hand-crafted representation including molecular descriptors, string-based notations, and image (Zeng et al., 2022). Besides the node- and graph-level methods which represent atoms as nodes and bonds as edges, fragment-based approaches explicitly modeling molecular substructures to learn higher-order semantics. Existing methods mostly generate fragments based on pre-defined knowledge or purely geometric structure and learn fragment representations through autoregressive processes (Zhang et al., 2021b; Jin et al., 2020; Rong et al., 2020). These methods treat fragments merely as graph tokenizer (Liu et al., 2024), decomposing molecules into various of reconstruction units without fully incorporating fragment-level semantics into the learned representation.

## 3 PRELIMINARIES

Given a graph $G = V, A, X$, where $V$ is the set of $N$ nodes (atoms), $A \in \mathbb{R}^{N \times N}$ is the adjacency matrix, and $X = [x_1, x_2, \cdots, x_N]^\top \in \mathbb{R}^{N \times D}$ is the node feature matrix. Each entry $A_{ij} \in 0, 1$ indicates the presence ($A_{ij} = 1$) or absence ($A_{ij} = 0$) of a chemical bond between atoms $i$ and $j$. Each node feature $x_i$ represents an atom's properties, encoding its individual characteristics.

### 3.1 GRAPH NEURAL NETWORK

Graph Neural Network relies on message passing to learn useful representations for node based on their neighbors. Given an input graph $G = \{V, A, X\}$, the node embedding is calculated by:

$$H^k = M(A, H^{(k-1)}; W^{(k)}) = \text{ReLU}(\omega(\widetilde{A})H^{(k-1)}W^{(k-1)}), \qquad (1)$$

where $\widetilde{A} = A + I$, $\widetilde{D} = \sum_j \widetilde{A}_{ij}$ and $W^{(k)} \in \mathbb{R}^{d \times d}$ is a trainable weight matrix. $\omega(\widetilde{A})$ is the normalization operator that represents the GNN's aggregation function. An additional readout function $R$ combines the final node embeddings into a graph embedding $H_G$, which is formalized as:

$$H_G = \text{READOUT}(H_v^k | v \in V), \qquad (2)$$

where $V$ is the node set and $k$ denotes the layer index. READOUT is a permutation-invariant pooling function, such as summation. The resulting graph representation $H_G$ captures global structural and node-level semantic information of the graph and can be used for various downstream tasks.

### 3.2 MASKED GRAPH MODELING

Masked Graph Modeling (MGM) aims to pre-train a graph encoder using component masking for downstream applications. Specifically, it masks out some components (e.g., atoms, bonds, and fragments) of the molecules and then trains the model to predict them given the remaining components. Given a graph $G = \{V, A, X\}$, the general learning objective of MGM can be formulated as:

$$L_{\text{MGM}} = -\mathbb{E}_{V_m \in V} \left[ \sum_{v \in V_m} \log p(\widetilde{V_m} \mid V \setminus V_m) \right], \qquad (3)$$

where $V_m$ denotes the masked components of graph $V$ and $V \setminus V_m$ are the remaining components.

## 4 METHODOLOGY

The fundamental architecture of the GraSPNet is illustrated in Figure 3. In general, the architecture is designed to predict the representations of multiple semantic target by leveraging the learned representations of a context graph with missing information. In this section, we detail the design and implementation choices for each component of the architecture.

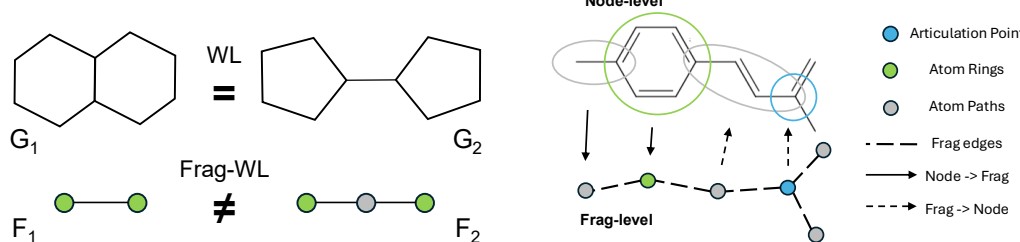

Figure 2: Illustration of our graph fragmentation process. The left figure shows graph $G_1$ and $G_2$ which can not be distinguished by WL test while higher-level fragment graph $F_1$ and $F_2$ exhibit different connections that can be distinguished by WL. The right figure shows an example of our graph fragmentation. The ring is first selected, followed by the extraction of multiple paths. The articulation points are designated as unique fragment types to prevent cycles in the fragment graph.

### 4.1 GRAPH FRAGMENTATION

Understanding data semantics is often considered essential in machine learning (Fei et al., 2023). Graph fragmentation methods aim to decompose an entire graph into structurally informative subgraphs that are closely related to the graph's properties. This hierarchical structure is crucial for enhancing the expressiveness of GNNS.

The widely used WL-test (Leman & Weisfeiler, 1968) which captures structural differences between graphs by repeatedly updating node labels based on their neighbors had been proved to be the expressiveness upper bound of message passing network (Xu et al., 2019). Higher-level fragmentation graph with nodes and fragments representation together with the interaction between them is more expressiveness and can be more powerful than 2-WL test in distinguish graph isomorphic. As illustrated in Figure 2, WL-test fails to distinguish between $G_1$ and $G_2$ where both graphs exhibit symmetric structures at the atom level, leading to identical label refinement through WL iterations. However, as for their corresponding fragment graphs, $F_1$ forms a simple two-node graph, while $F_2$ includes a three-node chain with a distinct central 'path' fragment. This structural asymmetry is detectable by the WL test on the fragment level, enabling better discrimination of molecular graphs.

Existing fragmentation methods are typically rule-based procedures, including BRICS (Degen et al., 2008) and RECAP (Lewell et al., 1998) which follow chemical heuristics but often generate large, sparse vocabularies with low-frequency or unique fragments, limiting generalization. Others, like METIS (Karypis & Kumar, 1998), use graph partitioning algorithm to minimize edge cuts may disrupt chemically meaningful structures and produce non-deterministic, molecule-specific fragments.

To construct a fragmented graph with high expressiveness and strong generalization ability, the fragmentation method should both capture key structural fragments and generalize across diverse molecular graphs. We propose a fragmentation strategy that decomposes each molecule into rings, paths and articulation points, forming a higher-level graph that supports learning both fragment representations and their structural relationships. Specifically, given the SMILES of a molecule, we transform it into graph with $G = \{V, A, X\}$ using RDKIT where $V$ and $A$ are atoms (nodes) and bonds (edges) respectively, $X$ is the corresponding atom feature. $V$ is then divided into subgraphs $V_1, \ldots, V_m$ where $m$ is the number of generated fragments using the following three steps. **Ring extraction.** We first identify all minimal rings in the molecular graph and form the first type of subgraphs, each corresponding to a ring. **Path extraction.** For the remaining nodes and edges, we extract paths in which all intermediate nodes have degree of 2, while the endpoints may have other degrees. These form the second type of subgraphs. **Articulation point extraction.** Finally, any remaining nodes with degree greater than 3 are selected as individual articulation-point subgraphs.

This process results in a set of overlapping subgraphs, where each node is assigned to exactly one subgraph, except for connector nodes that bridge two adjacent substructures. Figure 2 illustrates an example of our fragmentation method. This design preserves topological information and prevent most circles in the fragmentation graph, enabling effective fragment-level representation learning.

All fragments induce a new fragment node set $V^f$, with an associated adjacency matrix $A^f \in [0,1]^{m \times m}$ representing the connectivity between fragments and $A^{nf} \in [0,1]^{n \times m}$ mapping the orig-

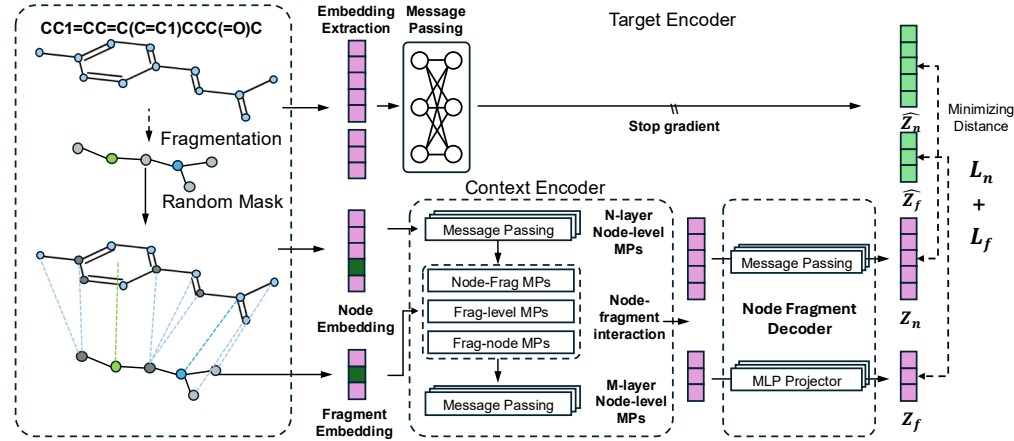

Figure 3: Overview of the Graph Semantic Predictive Network (GraSPNet) framework. The original molecule is fragmented to form a higher-level fragment graph. Masked node and fragment graphs are input into the context encoder, while the target encoder processes the original unmasked graphs. The predictor uses context representations to predict node and fragment embeddings, and the loss minimizes the distance between the prediction and the target encoder's representations.

inal node to fragments. Specifically, $\forall (i, j)$,

$$A_{ij}^f = 0 \;\Rightarrow\; V_i^f \cap V_j^f = \emptyset, \quad A_{ij}^f = 1 \;\Rightarrow\; V_i^f \cap V_j^f \neq \emptyset,$$

indicating whether two fragments share overlapping nodes. The size of the overlapping corresponds to the minimum cut value between the two fragment in the original graph.

## 4.2 MODEL

Based on the given graph with fragmentation, we proposed GraSPNet which is a predictive architecture focused on learning fragmentation level information. The model is constructed by nodes and fragmentation encoding, context encoder, target encoder, and predictor which is shown in Figure 3.

### 4.2.1 FRAGMENT ENCODING

We first introduce the graph information encoding which includes both node encoding and fragmentation encoding. Previous work generate fragment embeddings by aggregating nodes or one hot encoder. To achieve better generalization and ensure that similar fragmentations share similar initial features, we follow the (Wollschläger et al., 2024) by incorporating the fragment class and size into the embedding. Formally, the fragment embeddings are generated as:

$$h_f = W_1 \cdot X(f) \| \alpha \cdot (W_2 \cdot X(f)), \tag{4}$$

where $h_f$ is the fragment embedding, $W_1, W_2$ are two learnable encoding matrix, $X(f)$ is the one-hot initial vector representing fragment type, scaling factor $\alpha$ is equals to the fragment size.

### 4.2.2 CONTEXT ENCODER

We design our context encoder to leverage hierarchical representation learning. To train the encoder, a destructive data augmentation is applied by randomly masking nodes and fragments in the input graph. The masking process is modeled as a Bernoulli distribution applied independently to each nodes and fragments which is defined formally as follows:

$$\mathbf{V_m} \sim Bernoulli(p),$$

where $p < 1$ is the masking ratio. We denote the masked node set as $V_m$ and the remaining node set as $V - V_m$. The fragment masked set are define as $V_m^F$ with the same $p$.

The input node features $X^n \in \mathbb{R}^{d_n \times 1}$ are projected into $d$-dimensional embedding through a linear embedding layer:

$$h_i^n = W^n \cdot X_i^n + b,$$

where $W^n \in \mathbb{R}^{d \times d_n}$ and $b \in \mathbb{R}^d$ are learnable parameters. $h_f$ is generated from Equation 4.

Given the initial node and fragment embeddings, we apply a series of $L$ message-passing layers, where both node and fragment representations are iteratively updated using a graph neural network. The message-passing procedure in the encoder consists of four components: $M_{n \to n}$, $M_{n \to f}$, $M_{f \to f}$, and $M_{f \to n}$, which denote message passing from nodes to nodes, nodes to fragments, fragments to fragments, and fragments to nodes, respectively. Here, $M_{n \to n}$ corresponds to standard message passing over the original atom nodes, while the remaining components collectively form the fragment-level message passing layer.

For each layer, the node representations are first updated through node-level message passing. If a fragment layer is applied, the updated node features are then aggregated into the fragments to which each node belongs. Subsequently, higher-level fragment message passing is performed to capture structural information based on fragment representations and their connectivity. Finally, the updated fragment features are injected back into the corresponding node representations. The message passing between nodes and fragments can be represented as:

$$H_{\text{node}}^{(l)} = f_{\text{node}}^{(l)}\big((A_{\text{node}}, H_{\text{node}}^{(l-1)}), W_0^{(l)}\big), \quad H_{\text{frag}}^{(l)} = f_{\text{node-frag}}\big((A_{\text{node-frag}}, H_{\text{frag}}^{(l-1)}, H_{\text{node}}^{(l)}, W_1^{(l)}\big),$$

$$H_{\text{frag}}^{(l)} = f_{\text{frag}}\big((A_{\text{frag}}, \text{pool}(H_{\text{frag}}^{(l)})), W_2^{(l)}\big), \quad H_{\text{node}}^{(l)} = f_{\text{frag-node}}^{(l)}\big((A_{\text{frag-node}}, H_{\text{node}}^{(l)}, H_{\text{frag}}^{(l)}), W_3^{(l)}\big),$$

where $l$ is the layer index, $W_0, W_1, W_2, W_3 \in \mathbb{R}^{d \times d}$ are learnable weight matrices of the $l^{th}$ message passing layer with feature dimension $d$. Let $A_{\text{node}} \in \mathbb{R}^{n \times n}$ denote the original adjacency matrix of the molecular graph. The matrices $A_{\text{node-frag}}, A_{\text{frag-node}} \in \mathbb{R}^{n \times m}$ and $A_{\text{frag}} \in \mathbb{R}^{m \times m}$ represent the node-to-fragment mapping, fragment-to-node mapping, and the adjacency between fragments, respectively, where $n$ is the number of nodes and $m$ is the number of fragments. Function $f_{\text{node}}^{(l)}$ and $f_{\text{frag}}^{(l)}$ denotes the $l$-th node-level and fragment-level message-passing layer, which can be any standard GNN architecture (e.g., GCN, GIN). $f_{\text{frag-node}}$ denote a fragment-level message passing, implemented using an MLP and guided by the fragment connectivity and node-fragment associations. The aggregation from node features to fragment features is defined via a pooling function: $h_p = \frac{1}{|V_p|} \sum_{i \in V_p} h_{i,p}^l \in \mathbb{R}^d$ where $1 \le p \le m$ and $V_p$ is the set of nodes assigned to fragment $p$.

### 4.2.3 TARGET REPRESENTATION

The target encoder adopts the same architecture as the context encoder, incorporating both node and fragment encoding along with $k$ layers of message passing with fragment layer to learn representations of the original input graph enriched with higher-level semantic information. To prevent representation collapse, we use a lighter version of the context encoder with fewer layers and update its parameters using an Exponential Moving Average (EMA) of the context encoder parameters.

### 4.3 PREDICTOR AND LOSS

Given the learned node representations $Z_n$ and fragment representations $Z_f$ from the context encoder, the objective is to capture richer semantic information beyond individual node embeddings. Our prediction model consists of two components: node representation prediction and fragment representation prediction.

For node prediction, we first remove the representations of masked nodes from the context output. The remaining node embeddings are passed through a $k$-layer message passing network to predict the target node representations. Unlike a simple MLP decoder, this approach encourages the encoder to capture topological dependencies rather than merely reconstructing individual node features. The prediction is defined as: $\hat{Z}_n = g(A, Z_n^{context}; W)$, where $g$ denotes the message passing function and $A$ is the node adjacency matrix. For fragment prediction, a $k$-layer MLP is used to predict the target fragment embeddings from the context output: $\hat{Z}_f = MLP(Z_f^{context}; W)$.

Table 1: Evaluation on molecular property prediction tasks. For each downstream dataset, we report the mean and standard deviation of the ROC-AUC (%) scores over three random scaffold splits. The best and second best scores are marked bold and underline, respectively.

| | BBBP | Tox21 | ToxCast | Sider | MUV | HIV | Bace | Clintox |
|---|---|---|---|---|---|---|---|---|
| No Pre-train | 67.8±1.4 | 73.9±0.8 | 62.4±0.4 | 58.3±1.8 | 73.4±2.5 | 76.3±1.2 | 76.8±2.6 | 62.6±4.4 |
| MAE | 68.7±1.3 | 75.5±0.5 | 62.0±0.8 | 58.0±1.0 | 69.7±1.5 | 74.2±2.2 | 77.2±1.6 | 70.1±3.2 |
| Infomax | 69.2±0.7 | 74.6±0.5 | 61.8±0.8 | 60.1±0.7 | 74.8±1.5 | 75.0±1.3 | 76.3±1.8 | 71.2±2.5 |
| Attr mask | 65.6±0.9 | 77.2±0.4 | 63.3±0.8 | 59.6±0.7 | 74.7±1.9 | 77.9±1.2 | 78.3±1.1 | 77.5±3.1 |
| CP | 72.1±1.5 | 74.3±0.5 | 63.5±0.3 | 60.2±1.2 | 70.2±2.8 | 74.4±0.8 | 79.2±0.9 | 70.2±2.6 |
| ADGCL | 70.5±1.8 | 74.5±0.4 | 63.0±0.5 | 59.1±0.9 | 71.5±2.2 | 75.9±1.8 | 74.2±2.4 | 78.5±3.7 |
| GraphCL | 71.4±1.1 | 74.5±1.0 | 63.1±0.4 | 59.4±1.3 | 73.8±2.0 | 75.6±0.9 | 78.3±1.1 | 75.5±2.4 |
| JOAO | 71.8±1.0 | 74.1±0.8 | 63.9±0.4 | 60.8±0.6 | 74.2±1.2 | 76.2±1.8 | 77.2±1.7 | 79.6±1.4 |
| BGRL | 72.5±0.9 | 75.8±1.0 | 62.1±0.5 | 60.4±1.2 | 76.7±2.8 | 77.1±1.2 | 74.7±2.6 | 65.5±2.3 |
| GraphMAE | 71.7±0.8 | 76.0±0.9 | 64.8±0.6 | 60.0±1.0 | 76.3±1.9 | 75.9±1.8 | 81.7±1.6 | 80.5±2.0 |
| MGSSL | 69.7±0.9 | 76.5±0.3 | 64.1±0.7 | 61.5±1.0 | 76.3±1.9 | **79.5±1.8** | 79.7±1.6 | 80.7±2.1 |
| SimSGT | 72.8±0.5 | 76.8±0.9 | 65.2±0.8 | 60.6±0.8 | **79.0±1.4** | 77.6±1.9 | 81.5±1.0 | 82.0±2.6 |
| **GraSPNet** | **74.4±1.5** | **77.3±0.8** | **65.5±0.5** | **62.5±1.1** | 78.5±1.3 | 78.0±0.8 | **82.9±3.1** | **84.1±2.1** |

The loss function is a commonly used MSE loss to measure the distance between the target encoder fragment representation and the predicted fragment representation, which is formally written as:

$$L = \alpha \frac{1}{|V_m^n|} \sum_{i=1}^{|V_m^n|} D(\hat{Z}_n^i, Z_n^i) + (1 - \alpha) \frac{1}{|V_m^f|} \sum_{i=1}^{|V_m^f|} D(\hat{Z}_f^i, Z_f^i),$$

where $V_m^n$ and $V_m^f$ denote the masked node and fragment sets, respectively; $|V_m^n|$ and $|V_m^f|$ represent their cardinality, $D(\cdot)$ denotes the Euclidean distance between two vectors and $\alpha$ is hyperparameter.

## 5 EXPERIMENTS

In this section, we conduct extensive experiments to evaluate the performance of GraSPNet across various benchmark datasets, aiming to assess the model's expressiveness and generalization ability. We also conduct additional studies on the impact of fragment layer position on performance.

### 5.1 SETTINGS.

We use the open-source RDKit package to pre-process SMILES strings and perform fragmentation for various datasets. For pretraining GraSPNet and all baseline models, we leverage 2 million unlabeled molecules from the ZINC15 database (Sterling & Irwin, 2015). During downstream prediction, only the pre-trained context encoder is used and fine-tuned on each benchmark dataset. Final predictions are obtained by applying mean pooling over all node representations in a graph, followed by a linear projection layer. Detailed training configurations are provided in the Appendix.

### 5.2 EVALUATION AND METRICS

The evaluation focuses on molecular property prediction tasks using benchmark datasets from MoleculeNet (Wu et al., 2018), a collection compiled from multiple public databases. Specifically, we select eight classification datasets related to physiological and biophysical property prediction: BBBP, Tox21, ToxCast, SIDER, ClinTox, MUV, HIV, and BACE. These datasets cover a diverse range of molecular properties, including blood–brain barrier permeability (BBBP), toxicity prediction (Tox21, ToxCast, ClinTox), adverse drug reactions (SIDER), bio-activity against HIV (HIV), and binding affinity to drug targets (BACE). We also conduct experiment on regression task datasets including FreeSolv, ESOL and Lipophilicity which focus on physical chemistry. Detailed descriptions are provided in the Appendix. For downstream tasks, datasets are split 80/10/10% into training, validation, and test sets using the scaffold split protocol, in line with prior works.

### 5.3 BASELINES.

As part of our experimental baselines, we include well-established self-supervised methods from different categories: **contrastive-based** methods such as Infomax (Veličković et al., 2018), ADGCL (Pan et al., 2018), GraphCL (You et al., 2020), and JOAO (You et al., 2021); **generative-based** methods including MAE (Kipf & Welling, 2016b), ContextPred (Hu et al., 2020a), and GraphMAE (Hou et al., 2022); **predictive methods** like Attribute Masking (You et al., 2020), BGRL (Thakoor et al., 2021), and SimSGT (Liu et al., 2024); as well as **fragment-based** method, MGSSL (Zhang et al., 2021b), HiMOL (Zang et al., 2023) and S-CGIB (Lee et al., 2025). We provide a comparative evaluation of these baselines against our proposed method.

### 5.4 MOLECULE PROPERTY PREDICTION.

The molecule property prediction tasks follows the setting and we use the same node encoder structure with 5 layers of graph isomorphism network (GIN) along with batch normalization for each layer. The higher-level fragment message passing layer is constructed with 2 layers of GIN to avoid over-squashing in smaller fragment graph. Table 11 reports the ROC-AUC (%) scores for eight molecular property prediction benchmarks using different self-supervised pretraining strategies. Overall, GraSPNet consistently achieves the best or second-best performance across most tasks, demonstrating its strong generalization ability.

Across most datasets, pretraining helps significantly. Compared with the baseline without pretraining, all self-supervised methods improve ROC-AUC, highlighting the benefit of pretraining on molecular graphs. GraSPNet outperforms all competing methods on BBBP (74.4%), Tox21 (77.3%), ToxCast (65.5%), Sider (62.5%), Bace (82.9%), and Clintox (84.1%), indicating superior transferability to diverse biochemical endpoints. On MUV, SimSGT achieves the highest ROC-AUC (79.0%), followed closely by GraSPNet (78.5%). For HIV, MGSSL leads with 79.5%, while GraSP-Net is second-best (78.0%). Compared to earlier contrastive learning approaches such as GraphCL, JOAO, and ADGCL, GraSPNet yields clear improvements (e.g., +3–5% on BBBP and Clintox). These results suggest that GraSPNet better captures transferable molecular semantics, especially for small datasets like BBBP and Clintox where pretraining benefits are most pronounced. Furthermore, its robust performance across both toxicity-related (Tox21, Sider) and bioactivity-related (HIV, Bace) benchmarks indicates its adaptability across heterogeneous molecular tasks.

We further compare our method with other fragment-based approaches, as reported in Table 10. Our method achieves the best performance on Clintox (82.5) and MUV (78.5), and ranks second on HIV and BBBP. Compared with S-CGIB, which relies on aggregating one-hop subgraphs of each node for pre-training and fine-tuning, our approach generates meaningful fragmentations that enable more effective representation learning while avoiding the high memory overhead of subgraph enumeration. In contrast to MGSSL, which constructs fragments dynamically during training and thus incurs substantial computational cost, our method performs fragmentation in a more efficient manner, leading to faster training without sacrificing predictive accuracy.

Apart from classification tasks, we also reports fine-tuning performance on three regression benchmarks from physical chemistry as shown in Table 5 in Appendix, measured by RSE (lower is better). Our method achieves the best results on all three tasks. These results highlight GraSPNet's ability to capture higher-level molecular semantics that are critical for modeling fundamental physical chemistry properties such as solubility and lipophilicity.

Overall, these results demonstrate that our approach consistently improves predictive performance across multiple datasets, highlighting the strength of incorporating fragment-level message passing, while also identifying areas for future enhancement on specific molecular tasks.

### 5.5 ABLATION STUDY.

Table 2 presents an ablation study on four molecular property prediction datasets: clintox, bbbp, sider, and bace. The baseline GINE model performs moderately across all tasks. Removing fragment information during masked node reconstruction ("w/o F") results in noticeable performance drops on most datasets, highlighting the contribution of chemically meaningful fragments to representation learning. Excluding higher-level message passing ("w/o Higher-MP") further degrades performance, demonstrating the importance of multi-level interaction modeling. The full model,

Table 2: Ablation study on molecular prediction tasks. w/o F: without fragment information. w/o H-MP: without higher-level MP.

| Type | Clintox | BBBP | Sider | Bace |
|------|---------|------|-------|------|
| GINE | 71.8 | 70.2 | 59.6 | 76.9 |
| w/o F | 78.9 | 71.3 | 60.0 | 80.7 |
| w/o H-MP | 81.9 | 72.8 | 61.3 | 81.2 |
| Full | **84.1** | **74.2** | **62.5** | **82.9** |

Table 3: Comparison with other fragment-based methods on four molecular property prediction benchmarks.

| | Clintox | MUV | HIV | BBBP |
|------|---------|-----|-----|------|
| MGSSL | 80.7 | 76.3 | **79.5** | 69.7 |
| S-CGIB | 74.6 | 74.1 | 77.3 | **85.4** |
| HiMOL | 80.8 | 76.3 | 77.1 | 70.5 |
| Ours | **82.5** | **78.5** | 78.0 | 74.4 |

which incorporates both fragment-aware masking and hierarchical message passing, consistently achieves the best results across all tasks, indicating that both components are complementary and essential for effective molecular representation learning.

### 5.6 FRAGMENT LAYER ANALYSIS.

Since the fragmentation layer can be placed after any node-level message passing layer, we conduct experiments to test the influence of node-fragment interaction after each layer. The result is shown in Figure 4. We conduct pre-training on 20,000 molecules on ZINC and fine-tune it on 4 downstream tasks: clintox, bbbp, sider, and bace. We use 5-layer of GINE as context encoder and add the node-fragment interaction layer and higher-level fragment layer after each message passing layer. The result shows that as the depth increases from layer 1 to layer 3, the performance gradually improves on all datasets, indicating that early layer hierarchical message passing enriches node representations with meaningful local semantical information, which benefits multiple downstream performance. Placing the fragment layer after

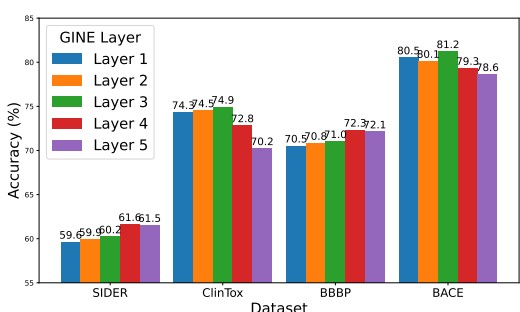

Figure 4: Performance of incorporating fragment information after different GINE layers in the context encoder.

layer 4 provides marginal gains for BBBP and SIDER but begins to degrade performance on Clin-Tox and BACE, and adding the fragment layer after layer 5 consistently results in performance deterioration across all tasks. This shows that late layer node and fragment information interaction may enlarge the influence of over-smoothing and be harmful to downstream performance. Overall, these results demonstrate that positioning the fragment layer at an intermediate depth achieves the best trade-off between enriched semantics and avoiding over-smoothing, thereby enhancing fragment-aware molecular representations.

## 6 CONCLUSION

In this work, we proposed a fragment-based pretraining framework for molecular graph representation learning that jointly predicts node and fragment embeddings to capture semantically meaningful graph representations. By introducing fragmentation based on both geometric and chemical structure, we construct higher-level graph abstractions that are expressive yet maintain a lower vocabulary size, promoting better generalization. The representation is learned through hierarchical message passing and an embedding prediction objective at both node and fragment levels, enabling the model to capture semantic information across multiple resolutions. We evaluate our approach on several benchmark datasets under transfer learning settings and demonstrate competitive performance. A limitation of our method is that the fragmentation design is tailored to molecular graphs and may not generalize well to node classification tasks, such as social networks. In future work, we plan to incorporate multimodal signals such as 3D molecular structures and natural language annotations to further explore the benefits of fragmentation in graph learning.

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

# A   DATASETS DESCRIPTION

The benchmark datasets used for downstream prediction tasks are summarized in Table 4, including their descriptions, number of graphs, and number of prediction tasks.

Table 4: Summary of molecular property prediction benchmarks.

| Dataset | Description | Number of Graphs | Number of Tasks |
|---|---|---|---|
| BBBP | Blood-brain barrier permeability | 2,039 | 1 |
| Tox21 | Toxicology on 12 biological targets | 7,831 | 12 |
| ToxCast | Toxicology via high-throughput screening | 8,575 | 617 |
| SIDER | Adverse drug reactions of marketed medicines | 1,427 | 27 |
| ClinTox | Clinical trial failures due to toxicity | 1,478 | 2 |
| MUV | Validation of virtual screening techniques | 93,087 | 17 |
| HIV | Ability to inhibit HIV replication | 41,127 | 1 |
| BACE | Inhibitors of human $\beta$-secretase 1 binding results | 1,513 | 1 |

# B   DISTRIBUTION OF FRAGMENTS

The distribution of fragment size and number of each fragments including rings, paths and articulation per graph in pretraining ZINC dataset are shown in Figure 6.

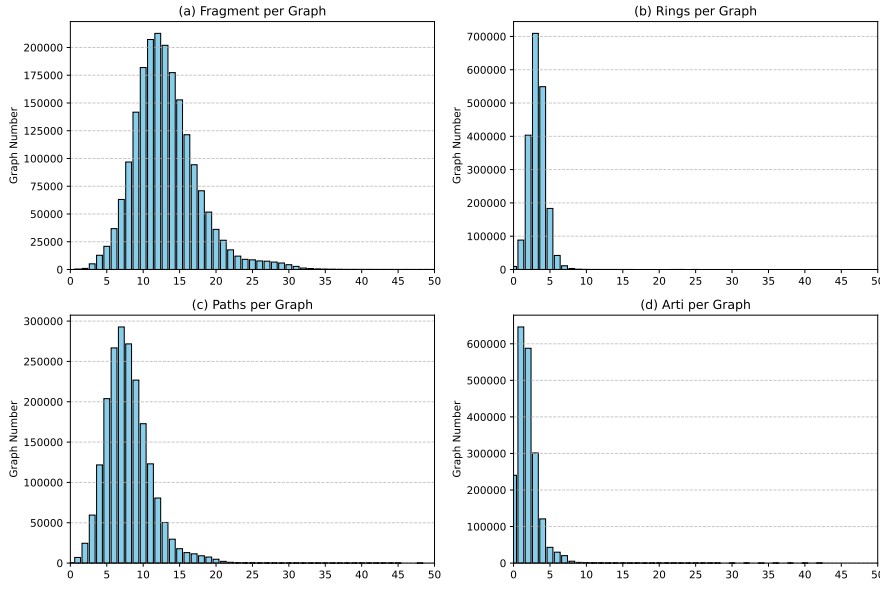

Figure 5: Distribution of structural components in molecular graphs. Each subplot shows the distribution of (a) fragment sizes, (b) number of rings, (c) number of paths, and (d) number of articulation points per graph. The x-axis represents the count of each structure, and the y-axis shows the number of graphs with that count.

# C   ADDITIONAL EXPERIMENTS

In this section we list the additional experimental result which is mentioned in the main paper. Table 5 shows the RMSE of regression tasks.

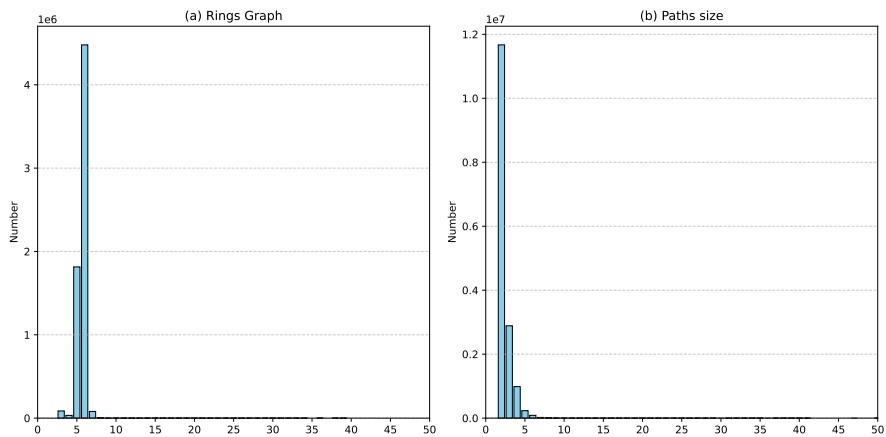

Figure 6: Distribution of (a) Rings and (b) Paths size in ZINC dataset.

Table 5: Performance comparison on regression tasks in terms of RMSE($\downarrow$).

| Methods | FreeSolv | ESOL | Lipophilicity |
|---|---|---|---|
| ContextPred | $3.195 \pm 0.058$ | $2.190 \pm 0.026$ | $1.053 \pm 0.048$ |
| AttrMasking | $4.023 \pm 0.039$ | $2.954 \pm 0.087$ | $0.982 \pm 0.052$ |
| EdgePred | $3.192 \pm 0.023$ | $2.368 \pm 0.070$ | $1.085 \pm 0.061$ |
| Infomax | $3.033 \pm 0.026$ | $2.953 \pm 0.049$ | $0.970 \pm 0.023$ |
| JOAO | $3.282 \pm 0.002$ | $1.978 \pm 0.029$ | $1.093 \pm 0.097$ |
| GraphCL | $3.166 \pm 0.027$ | $1.390 \pm 0.363$ | $1.014 \pm 0.018$ |
| GraphFP | $2.528 \pm 0.016$ | $2.136 \pm 0.096$ | $1.371 \pm 0.058$ |
| MGSSL | $2.940 \pm 0.051$ | $2.936 \pm 0.071$ | $1.106 \pm 0.077$ |
| GROVE | $2.712 \pm 0.327$ | $1.237 \pm 0.403$ | $0.823 \pm 0.027$ |
| SimSGT | $1.953 \pm 0.038$ | $1.213 \pm 0.032$ | $0.835 \pm 0.037$ |
| **GraSPNet(Ours)** | $\mathbf{1.232 \pm 0.05}$ | $\mathbf{1.161 \pm 0.37}$ | $\mathbf{0.813 \pm 0.052}$ |

## D    FURTHER RELATED WORK

**Representation Learning on Molecules.** Representation learning on molecules has made use of hand-crafted representation including molecular descriptors, string-based notations, and image (Zeng et al., 2022). Graph-based representation learning are currently the state-of-the-art methods as it can capture geometric information in molecule structure. In this setting, molecules are typically modeled as 2D graphs, where atoms are represented as nodes and bonds as edges, with associated feature vectors encoding atom and bond types (Hu et al., 2020b). GNN pretraining are commonly used for molecule representation learning using contrastive learning (You et al., 2020; 2021; Suresh et al., 2021; Xu et al., 2021), auto-encoding (Hou et al., 2022; 2023), masked component modeling (Xia et al., 2023; Liu et al., 2024), or denoising (Zaidi et al., 2022; Liu et al., 2022). At the pretraining level, methods can be categorized into node-level, graph-level, and more recently fragment-level (Guo et al., 2023). Node-level methods capture chemical information at the atomic scale but are limited in representing higher-order molecular semantics, while graph-level methods may overlook fine-grained structural details.

**Joint embedding predictive architecture.** Joint-Embedding Predictive Architectures (LeCun, 2022; Garrido et al., 2024) are a recently proposed self-supervised learning architecture which combine the idea of both generative and contrastive learning methods. It is designed to capture the high-level dependencies between the input $x$ and the prediction object $y$ through predicting missing information in an abstract representation space. The JEPA framework has been implemented for images (Garrido et al., 2024), videos (Bardes et al., 2023) and audio (Fei et al., 2023) and shows a superior performance on multiple downstream tasks. It is claimed that JEPA can improve the

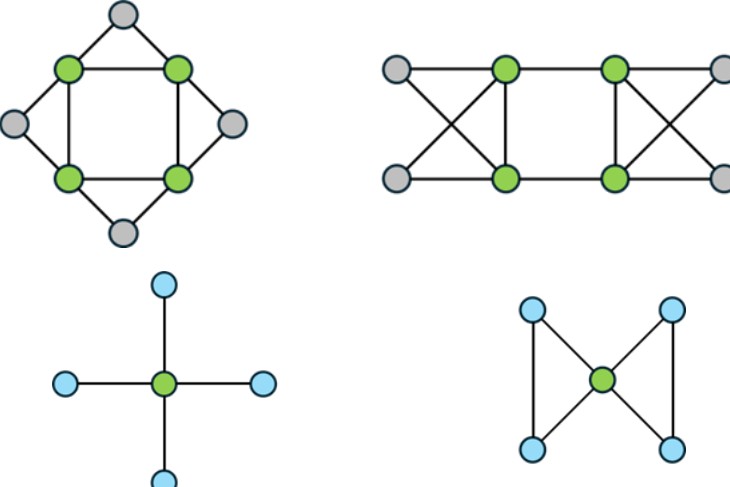

Figure 7: $G_1$ (top left) and graph $G_2$ (top right) with their corresponding higher-level graph on bottom left and bottom right. Graph $G_1$ and graph $G_2$ that are indistinguishable by 1-WL but distinguishable by Fragment-WL with higher-level graph.

semantic level of self-supervised representations without extra prior knowledge encoded (Assran et al., 2023). However previous work mostly focus on downstream tasks performance without actually provide evidence to support the semantic level comparison. For molecule graph learning, the natural properties of molecules allow us to evaluate the semantic information contained in the representation by detecting specific functional groups detection. This allows us to analyze and compare the semantic level of learned representation.

## E   LLM USAGE STATEMENT

We used large language model (LLM) solely for grammar checking and minor language editing. No part of the research ideation, methodology, analysis, or writing of scientific content relied on LLMs.

## F   REPRODUCIBILITY

To ensure reproducibility of our results, we provide the hyperparameter configurations used in both the self-supervised pretraining and downstream fine-tuning stages. Table 6 lists the key architectural choices, optimization settings, and training details consistently applied across experiments. The code will be release upon acception.

Table 6: Hyperparameters for Experiments in Self-supervised learning and fine-tuning

| Setting | Self-supervised Learning | Fine-tuning |
|---|---|---|
| Backbone GNN Type | GIN | GIN |
| Context Layer | 5 | 5 |
| Target Layer | 1 | 1 |
| PE type | None | None |
| Backbone Neurons | [300] | [300] |
| Fragment layer | [2,3] | [2,3] |
| Batch size | {32, 64, 128} | {32} |
| Fragment GNN Type | GIN | GIN |
| Projector Neurons | [300, 300] | [300, 300] |
| Pooling Layer | Global Mean Pool | Global Mean Pool |
| Learning Rate | {0.0001} | {0.0001, 0.005, 0.001} |
| EMA | {0.996, 1.0} | {None} |
| Masking Ratio | {0.35} | {0} |
| Training Epochs | {100} | {100} |

Table 7: Ablation study on articulation point.

| Type | Clintox | BBBP | Sider | Bace | Mean |
|---|---|---|---|---|---|
| w/o Arti | 70.16 | 66.84 | 58.14 | **77.81** | 68.24 |
| Full | **70.35** | **67.30** | **59.61** | 76.86 | **68.53** |

Table 8: Training time.

| Model | GraphMAE | Mole-BERT | S2GAE | GraphMAE2 | SimSGT | GraSPNet |
|---|---|---|---|---|---|---|
| Pretrain Time | 527 min | 2199 min | 1763 min | 1195 min | 645 min | 769 min |

Table 9: RDKiT fragments prediction accuracy of GraSPNet.

| Fragment | epoxide | lactam | morpholine | oxazole | tetrazole | NO | ether | furan | guanido | halogen | piperdine |
|---|---|---|---|---|---|---|---|---|---|---|---|
| Acc | 99.5 | 99.95 | 99.45 | 99.10 | 98.85 | 99.60 | 91.75 | 99.60 | 99.55 | 94.35 | 95.70 |

Table 10: RDKiT fragments prediction accuracy of GraSPNet.

| Fragment | thiazole | thiophene | urea | allylic oxid | amide | amidine | azo | benzene | imidazole | imide | piperzine | pyridine |
|---|---|---|---|---|---|---|---|---|---|---|---|---|
| Acc | 99.05 | 99.70 | 99.75 | 95.45 | 94.65 | 99.65 | 99.80 | 92.25 | 98.75 | 99.4 | 97.2 | 97.15 |

Table 11: More baselines.

| | BBBP | Tox21 | ToxCast | Sider | MUV | HIV | Bace | Clintox |
|---|---|---|---|---|---|---|---|---|
| GraphLOG | 67.2±1.3 | 76.0±0.8 | 63.6±0.7 | 59.8±2.1 | 72.8±1.8 | 72.5±1.6 | 82.8±0.9 | 76.9±1.9 |
| S2GAE | 67.6±2.0 | 69.6±1.3 | 58.7±0.8 | 55.4±1.3 | 60.1±2.4 | 68.0±3.8 | 68.6±2.1 | 59.6±1.1 |
| Mole-BERT | 70.8±0.5 | 76.6±0.7 | 63.7±0.5 | 59.2±1.1 | 77.2±1.1 | 76.5±0.8 | 82.8±1.4 | 77.2±1.4 |
| GraphMAE2 | 71.6±1.6 | 75.9±0.8 | 65.6±0.7 | 59.6±0.6 | **78.5±1.1** | 76.15±2.2 | 81.0±1.4 | 78.8±3.0 |
| **GraSPNet** | **74.4±1.5** | **77.3±0.8** | **65.5±0.5** | **62.5±1.1** | **78.5±1.3** | **78.0± 0.8** | **82.9±3.1** | **84.1±2.1** |

