# OpenReview forum: "Hierarchical Molecular Representation Learning via Fragment-Based Self-Supervised Embedding Prediction"
_ICLR.cc/2026/Conference — Submitted to ICLR 2026_

### Official Review · Reviewer_Um5C · 2025-10-18

**Soundness:** 3
**Presentation:** 2
**Contribution:** 2
**Rating:** 4
**Confidence:** 4

**Summary:**

The authors present GraSPNet (Graph Semantic Predictive Network), a hierarchical SSL framework for molecular representation learning, modeling semantically meaningful molecular fragments, allowing the network to capture hierarchical and chemically relevant substructures without relying on a predefined vocabulary.
It first decomposes molecular graphs into fragments, encodes them using message-passing GNNs, and models interactions between nodes and fragments through node-fragment and fragment-fragment message passing. GraSPNet employs masked embedding prediction at both node and fragment levels to jointly capture fine-grained and high-level semantic dependencies.

**Strengths:**

- The introduction of both node- and fragment-level prediction enables multi-resolution representation learning, which is biologically and chemically intuitive.

- To avoid predefined chemical substructures makes the approach flexible and domain-agnostic, Self-Supervised and Vocabulary-Free allow it to generalize across molecular datasets.

**Weaknesses:**

- The idea of leveraging molecular fragments for hierarchical or semantic representation learning is not entirely new. Recent studies, such as GraphFG, S-CGIB, and other fragment- or motif-based molecular pretraining methods, have already explored similar directions.

- While the empirical results are strong, there is no theoretical discussion about the expressive power of the proposed hierarchical GNN relative to existing architectures (e.g., WL hierarchy).

- Both node- and fragment-level masked prediction could lead to overlapping learning signals; an ablation study is needed to disentangle their respective contributions.

**Questions:**

- From a theoretical perspective, does GraSPNet offer higher expressive power than 1-WL GNNs? For instance, can fragment-level reasoning distinguish certain non-isomorphic molecular structures that node-level GNNs cannot?

- How GraSPNet captures chemically meaningful substructures?

- To me, the prediction task is somehow not related to the molecular graph structure preservation, as they are from different perspectives. It is difficult to say that the model could transfer the correct patterns well to downstream datasets. Is there any theoretical analysis or proof that a prediction task is sufficient for an SSL task (in both structural and semantic preservation)?

---

> ### Author Response · Authors · 2025-11-26
>
> We thank the reviewer for the thoughtful comments and detailed feedback. We address each point below.
>
> # Response to Q1 and Weakness 2
>
> Yes, from a theoretical design perspective, GraSPNet is more expressive than standard 1-WL GNNs. Standard Message Passing Neural Networks (MPNNs) are bounded by the expressiveness of the 1-Weisfeiler-Lehman (1-WL) test, meaning they cannot distinguish between graphs that are isomorphic at the local neighborhood level but distinct globally. GraSPNet explicitly overcomes this limitation by introducing a hierarchical fragment graph. As detailed in Section 4.1, the method decomposes the graph into rings, paths, and articulation points to form a higher-level graph $F$. We know that If two graph G1 = (V1, E1,X1), G2 = (V2, E2,X2) have the same node features X1 = X2 and all nodes with the same node feature have the same neighborhood, they are indistinguishable by WL.
>
> So consider a general 3-node cycle distinguishing problem, *two graphs G1 and G2 with 8 nodes and 12 edges each* are shown in the **Appendix Figure 7**. G1 and G2 are non-isomorphic structures that cannot be distinguished by 1-WL since nodes with the same feature have the same neighborhood, every node is assigned the same additional node feature through the iteration which holds no additional information to distinguish the two graphs.
> The higher-level graph with both higher-level nodes and higher-level edges can inject additional information to distinguish the two graphs. In G1 3-cycle representation is connected to only one 4-cycle representation. In G2, 3-cycle representation is connected to one other 3-cycle representation and one 4-cycle representation. Hence, the fragment representation differs and distinguishes the two graphs.
>
> # Response to Q2 and Weakness 1
>
> GraSPNet captures meaningful substructures through a two-step process involving explicit decomposition and hierarchical message passing:
>
> - **Explicit Geometric/Chemical Decomposition:**
> Instead of learning fragments from scratch (which is hard) or using a fixed vocabulary (which doesn't generalize), GraSPNet uses a deterministic algorithm (RDKit-based) to break the molecule into three specific types of chemically relevant subgraphs:
>     - Rings: Identifies minimal rings (e.g., Benzene).
>     - Paths: Extracts connecting chains where nodes have a degree of 2.
>     - Articulation Points: Identifies high-degree junction atoms.
>     - Note: These fragments are embedded not just by identity, but by their constituent atom features and size.
> - **Hierarchical Message Passing & Prediction:**
> The "capturing" happens via the specific message passing flow defined in the context encoder section on page 6:
>     - Node to Fragment: Atoms send features to the fragments they belong to.
>     - Fragment to Fragment: Fragments exchange information to understand the global skeleton of the molecule.
>     - Fragment to Node: The updated fragment semantics are injected back into the atoms.
>     - The Training Objective: By masking a fragment and forcing the model to predict its embedding based on the surrounding context (using the Target Encoder), the model is forced to learn the chemical rules that govern which substructures are valid in a specific context.
>
> As shown in Appendix B (Figure 5 & 6), we verify the chemical expressiveness of the learned representation of GraSPNet. Our extracted fragments follow natural distributions (e.g., 5- and 6-membered rings), confirming they capture chemically meaningful motifs rather than random graph clusters. We also conduct experiment to further explore the chemical representativeness of GraSPNet generated molecule representation following the same evaluation framework of MOLGRAPHEVAL[1]. We conduct probing task to predict fragment counts for the 23 common RDKiT fragments using the frozen GraSPNet generated molecule representation and report the accuracy for each fragment.
>
> **Table: Accuracy of GraSPNet in predicting counts of 23 common RDKit fragments (Part 1).**
>
> | RDKit Fragment | epoxide | lactam | morpholine | oxazole | tetrazole | NO | ether | furan | guanido | halogen | piperdine |
> |----------------|---------|--------|-------------|---------|-----------|----|--------|--------|----------|----------|------------|
> | **Accuracy (%)** | 99.5 | 99.95 | 99.45 | 99.10 | 98.85 | 99.60 | 91.75 | 99.60 | 99.55 | 94.35 | 95.70 |
>
> **Table: Accuracy of GraSPNet in predicting counts of 23 common RDKit fragments (Part 2).**
>
> | RDKit Fragment | thiazole | thiophene | urea | allylic oxid | amide | amidine | azo | benzene | imidazole | imide | piperzine | pyridine |
> |----------------|----------|-----------|-------|--------------|--------|----------|------|----------|------------|--------|------------|-----------|
> | **Accuracy (%)** | 99.05 | 99.70 | 99.75 | 95.45 | 94.65 | 99.65 | 99.80 | 92.25 | 98.75 | 99.40 | 97.20 | 97.15 |
>
> [1] Wang, Hanchen, et al. "Evaluating self-supervised learning for molecular graph embeddings." NeurIPS (2023)

---

> ### Author Response · Authors · 2025-11-26
>
> # Response to Weakness 3
>
> We performed the exact ablation study in Section 5.5 (Table 2).
> - Results:
>     - w/o Fragment Prediction (w/o F): Performance drops significantly
>     - w/o Higher-level MP (w/o H-MP): Performance also drops.
>     - Full Model: Achieves the best performance.
>     - Interpretation: Node-level prediction focuses on local atomic valence and immediate neighbors (fine-grained), while fragment-level prediction focuses on functional groups and scaffold arrangements (coarse-grained).
>
> # Response to Q3
>
> We thank the reviewer for raising this fundamental concern regarding the sufficiency of our prediction task for structural and semantic preservation.
>
> Our objective is inspired by the Joint-Embedding Predictive Architecture (JEPA), which predicts the embedding of a target block from the masked context embedding[1]. Our model use node and fragment mask embedding prediction which:
>
> - **Focus on Semantics:** Unlike reconstruction methods (e.g. MAE) that are forced to model node-level detail of the masked region, our method is trained to predict the compressed, higher-level representation generated from the target encoder. This forces the network to learn both the node-level and chemically valid semantics (fragments) that are necessary for accurate downstream prediction.
>
> - **Information Maximization (InfoMax):** By minimizing the distance between $P(Z_{context})$ and $Z_{target}$, inherently maximizes the Mutual Information between the context and the target fragment's representation. To successfully maximize this information, the encoder must learn the conditional probability distribution of chemical structures and effectively 'understand' the relation between nodes and fragments through the GNN decoder.
>
> Empirical structural preservation experiment are provided in the Response to Q2. The high performance in identifying these specific, structure-defining motifs confirms that GraSPNet successfully captured the fine-grained structural and semantic dependencies required for chemical pattern recognition. This explain and validate its ability to transfer useful knowledge to downstream tasks.
>
> I hope these explanations resolve your concern! Please let us know if you have any remaining questions or require further clarification on any of these points.
>
> [1]Assran, Mahmoud, et al. "Self-supervised learning from images with a joint-embedding predictive architecture." Proceedings of the IEEE/CVF Conference on Computer Vision and Pattern Recognition. 2023.

---

### Official Review · Reviewer_rAXs · 2025-10-30

**Soundness:** 2
**Presentation:** 3
**Contribution:** 2
**Rating:** 4
**Confidence:** 4

**Summary:**

This paper introduces GraSPNet (Graph Semantic Predictive Network), a framework that addresses the neglect of chemical substructures in existing graph self-supervised learning methods. It uses a fragmentation technique to decompose molecules into rings, paths, and articulation points, constructing a multi-level graph structure. GraSPNet employs a self-supervised task similar to MAE, masking nodes and fragments, and predicts their embeddings using a context encoder.

**Strengths:**

1. The model achieves state-of-the-art or near-state-of-the-art performance on several challenging molecular property prediction benchmarks, particularly in transfer learning settings, demonstrating the effectiveness of its pretraining strategy and strong generalization ability.

2. The model explicitly models information transfer between atom-fragment and fragment-fragment, enabling it to capture higher-level chemical semantics that standard GNNs may overlook.

**Weaknesses:**

1. The fragmentation strategy is a fixed decomposition method based on heuristic rules. It remains unclear whether this decomposition approach is optimal for all downstream tasks. For example, some tasks may require substructure partitions with different granularities or types. Simply applying this method of partitioning could potentially disrupt the information carried within the molecular graph.

2. The "fragment-based" approach is not novel; utilizing substructures or motifs to enhance molecular graph representation learning has long been an established research direction in cheminformatics and graph machine learning.

3. Masking data at the fragment level directly could potentially disrupt the semantics represented by the data. After all, the premise of contrastive learning is to ensure that the semantics to be learned in the positive samples remain unchanged.

**Questions:**

1. Could you explain whether the fragmentation approach used is reasonable and preserves the molecular property features?

---

> ### Author Response · Authors · 2025-11-26
>
> We appreciate the reviewer’s positive assessment of GraSPNet’s performance and transfer learning capabilities. We value your critique regarding the fragmentation strategy and masking mechanism. Below, we address your concerns and clarify the theoretical and chemical rationale behind our design.
>
> # Response to Weakness 1 & Q1
>
> From the theoretical perspective, our fragmentation approach is strictly more expressive than node-level message passing. As explained in our response to Reviewer 4, introducing higher-level nodes and higher-level edges provides additional distinguishing information, making the resulting representation more expressive than 1-WL and using higher-level nodes alone.
> From the geometric perspective, our method is **grounded in graph bi-connectivity**. In graph theory, biconnected components (blocks) are maximal subgraphs that cannot be disconnected by removing a single vertex. Our extraction procedure mirrors the Block-Cut Tree decomposition: Rings correspond to biconnected components (cyclic blocks). Paths correspond to bridges or chains connecting these blocks. Articulation Points correspond to cut vertices. Recent work has shown that decomposing graphs into biconnected components provides a principled way to enhance GNN expressiveness beyond the 1-WL test. By explicitly modeling these stable substructures and their articulation points, we preserve the topological integrity of the graph better than arbitrary partitioning.
>
> From the chemical perspective, a chemically meaningful substructure is a set of atoms and bonds forming a stable and interpretable motif known to influence molecular properties. Typical motifs include aromatic and heterocyclic rings, aliphatic chains, branching points, fused ring systems, and common functional groups (e.g., –OH, –NH₂, –COOH). Chemically, our decomposition aligns with how molecules are constructed. Rings capture stable scaffolds (e.g., Benzene, Pyridine). Paths capture aliphatic linkers (e.g., alkyl chains) that determine flexibility. Articulation Points capture crucial junction atoms that determine 3D orientation. As shown in **Appendix B (Figure 5 & 6)**, our extracted fragments follow natural distributions (e.g., 5- and 6-membered rings), confirming they capture chemically meaningful motifs rather than random graph clusters. We also conduct experiment to further explore the chemical representativeness of GraSPNet generated molecule representation following the same evaluation framework of MOLGRAPHEVAL[3]. We conduct probing task to predict fragment counts for the 23 common RDKiT fragments using the frozen GraSPNet generated molecule representation and report the accuracy for each fragment.
>
> **Table: Accuracy of GraSPNet in predicting counts of 23 common RDKit fragments (Part 1).**
>
> | RDKit Fragment | epoxide | lactam | morpholine | oxazole | tetrazole | NO | ether | furan | guanido | halogen | piperdine |
> |----------------|---------|--------|-------------|---------|-----------|----|--------|--------|----------|----------|------------|
> | **Accuracy (%)** | 99.5 | 99.95 | 99.45 | 99.10 | 98.85 | 99.60 | 91.75 | 99.60 | 99.55 | 94.35 | 95.70 |
>
> **Table: Accuracy of GraSPNet in predicting counts of 23 common RDKit fragments (Part 2).**
>
> | RDKit Fragment | thiazole | thiophene | urea | allylic oxid | amide | amidine | azo | benzene | imidazole | imide | piperzine | pyridine |
> |----------------|----------|-----------|-------|--------------|--------|----------|------|----------|------------|--------|------------|-----------|
> | **Accuracy (%)** | 99.05 | 99.70 | 99.75 | 95.45 | 94.65 | 99.65 | 99.80 | 92.25 | 98.75 | 99.40 | 97.20 | 97.15 |
>
> So overall our method is both **geometrically** and **chemically** reasonable for preserving molecular and structural features. We also conduct experiments to show the effectiveness of our method.
>
> [1] Zhang, Bohang, et al. "Rethinking the Expressive Power of GNNs via Graph Biconnectivity." The Eleventh International Conference on Learning Representations.
>
> [2] Bonchev, Danail D., and O. G. Mekenyan, eds. Graph theoretical approaches to chemical reactivity. Vol. 9. Springer Science & Business Media, 2012.
>
> [3] Wang, Hanchen, et al. "Evaluating self-supervised learning for molecular graph embeddings." NeurIPS (2023)

---

> ### Author Response · Authors · 2025-11-26
>
> # Response to Weakness 2
>
> We acknowledge that utilizing substructures is an established direction. However, GraSPNet introduces two specific novelties that distinguish it from prior fragment-based works (e.g., MGSSL, molecular junctions):
>
> - **Hierarchical Interaction**: Most prior methods process fragments and atoms separately or simply pool atoms into fragments for a final readout. GraSPNet employs bidirectional hierarchical message passing (atom to fragment, fragment to fragment, fragment to atom). This allows the model to reason about the higher-level molecular semantic and inject that context back into local atoms during the encoding process without large vocabulary pool or time-consuming in-training fragments tree searching.
> - **Predictive vs. Contrastive/Generative**: GraSPNet moves beyond standard contrastive learning (which requires complex view augmentation) or reconstruction (which focuses on pixel/atom-level details). We utilize a Masked Embedding Prediction objective. By predicting the latent representation of missing fragments using the context of the remaining structure, the model is forced to learn the underlying chemical rules (e.g., "a benzene ring is likely here given the surrounding context") rather than just memorizing topology.
>
>
> # Response to Weakness 3
>
> The reviewer raises a valid concern regarding contrastive learning. However, we respectfully clarify that GraSPNet is not a contrastive learning method (which relies on invariant views of the same instance). Instead, it follows the Masked Graph Modeling (MGM) paradigm, similar to MAE in CV.
>
> In MGM, the goal is not to keep the positive samples unchanged, but to reconstruct the missing information from the context. Masking a fragment does temporarily "disrupt" the input, but this is the mechanism of learning: the Context Encoder is forced (by the mask and learning objective) to **understand the remaining molecular graph** well enough to get the missing node's and fragment's embedding. We use the Mask embedding prediction framework which predicts the missing embedding through context encoder + target encoder + predictor.
>
> The semantics are also not lost, they are distributed across the context. If the model successfully minimizes the prediction loss, it proves that it has learned to preserve and utilize the semantic dependencies between the visible context and the masked target. This "disruption" forces the model to learn robust, structure-aware representations that generalize well, as evidenced by our performance on downstream tasks.

---

### Official Review · Reviewer_NHXX · 2025-10-31

**Soundness:** 2
**Presentation:** 3
**Contribution:** 2
**Rating:** 2
**Confidence:** 4

**Summary:**

This paper presents GraSPNet, a hierarchical self-supervised framework for molecular graphs. It decomposes molecules into rings, paths, and articulation points as semantic fragments and jointly predicts node- and fragment-level embeddings through dual-channel message passing (node→fragment and fragment→fragment). Experiments on MoleculeNet benchmarks show consistent gains over prior self-supervised methods.

**Strengths:**

1. **Programmatic Fragmentation Strategy:** The model partitions molecules into structural subgraphs (rings, paths, articulation points) through a deterministic graph algorithm, ensuring reproducibility without predefined vocabularies.

2. **Hierarchical Message Passing:** The dual-channel design captures both local atomic and global fragment semantics.

3. **Comprehensive Experiments:** Evaluated on 8 classification and 3 regression benchmarks, showing consistent performance gains over GraphCL, GraphMAE, and MGSSL.

**Weaknesses:**

1. **Limited Novelty:** The core idea of hierarchical molecular representation has been explored in [1][2][3], and similar fragment-based or hierarchical pretraining exists. GraSPNet mainly integrates known techniques (fragment-level modeling + masked prediction + hierarchical GNN).

2. **Heuristic Fragment Extraction:** Although the method avoids chemical vocabularies, it still depends on hand-crafted structural heuristics (rings, paths, articulation points). A comparison with functional groups [4] or principal subgraph mining [5] is missing.

3. **Insufficient Analysis of Chemical Validity:** There is no visualization or quantitative evidence showing that extracted fragments correspond to meaningful chemical motifs.

References

[1] Li, Yuquan. Learning Hierarchical Interaction for Accurate Molecular Property Prediction. (2025).
[2] Jin, Wengong, Regina Barzilay, and Tommi Jaakkola. Hierarchical Generation of Molecular Graphs Using Structural Motifs. ICML, 2020.
[3] Luong, Kha-Dinh, and Ambuj K. Singh. Fragment-Based Pretraining and Finetuning on Molecular Graphs. NeurIPS 36 (2023): 17584–17601.
[4] Chen, Fangying, Junyoung Park, and Jinkyoo Park. A Molecular Hyper-Message Passing Network with Functional Group Information. arXiv:2106.01028 (2021).
[5] Kong, Xiangzhe, et al. Molecule Generation by Principal Subgraph Mining and Assembling. NeurIPS 35 (2022): 2550–2563.
[6] Zhang, Yikun, et al. Atomas: Hierarchical Adaptive Alignment on Molecule-Text for Unified Molecule Understanding and Generation. ICLR 2025.

**Questions:**

1. Could the authors integrate a **learned principal-subgraph** extraction mechanism (as in PS-VAE [5]) instead of fixed heuristics to enhance adaptability?

2. How does GraSPNet’s hierarchy differ from Atomas [6], which also performs automatic atom→fragment→molecule decomposition?

3. Have the authors analyzed the **distribution or diversity** of extracted fragments to ensure chemical representativeness?

---

> ### Author Response · Authors · 2025-11-26
>
> Thanks the reviewer for detailed assessment and for highlighting our rigorous evaluation and the reproducibility of our fragmentation strategy. We appreciate the opportunity to clarify our contributions regarding novelty, chemical validity, and the distinction from recent works like Atomas.
>
> # Response to Weakness 2 & Q1
> Incorporating a learned principal-subgraph extraction mechanism, such as the approach in PS-VAE, is indeed an interesting direction and we might go deeper into it in the future. However, in this work we deliberately chose a deterministic geometric strategy for three key reasons:
>
> - **Computational Efficiency:** Learned subgraph mining (e.g., frequent subgraph mining or PS-VAE) is computationally intensive and often creates memory bottlenecks due to the need for large vocabulary tables. The procedure of generating subgraphs during training (including methods like MGSSL) also increases the training time. Our decomposition method is deterministic, offline, and highly efficient, allowing GraSPNet to scale to millions of molecules (ZINC15) with minimal training overhead (see Appendix G).
>
> - **Completeness:** Functional group-based methods or principal subgraphs often leave "uninteresting" parts of the molecule (like aliphatic linkers) undefined or aggregates them into a "misc" bin. Our Rings+Paths+Articulation strategy ensures 100% atom coverage where every atom is assigned to a meaningful structural context.
>
> - **Generalization:** Learned vocabulary often suffers from long-tail problems and struggles to generalize to out-of-distribution molecules containing rare motifs. Our geometric rules (rings/paths) are chemically universal, ensuring the model can handle novel molecules without suffering from the "out-of-vocabulary" or long-tail problem.
>
> # Response to Weakness 1 & Q2
>
> While we acknowledge that hierarchical representation learning is an active area of research, GraSPNet distinguishes itself through the integration of explicit fragment-level message passing and a joint-embedding predictive objective.
>
> As for the difference between our method and Atomas[6].
>
> - First, Atomas jointly learns representations from SMILES and accompanying text, addressing the challenge of aligning local textual descriptions with molecular substructures without requiring explicit annotations. In contrast, our method operates purely on molecular graphs and does not rely on external text, avoiding the introduction of information beyond the molecular structure itself.
>
> - Second, Atomas is designed primarily to optimize aligned representations for generative tasks, whereas our work focuses on designing fragmentation methods and hierarchical pretraining methods on structure molecule data for robust molecular representation learning.
>
> - Third, the hierarchical alignment in Atomas is constructed across three levels—atoms, functional groups (motifs), and molecules—each aligned with corresponding levels of textual information. This introduces an additional source of supervision not available in raw SMILES. While the atom → fragment → molecule hierarchy is common in graph-based learning and consistent with the notion of “hierarchy,” the key difference lies in how the intermediate (fragment-level) hierarchy is **defined** and **utilized** during learning. Our approach constructs a higher-level graph that explicitly models information flow between subgraphs. This allows interaction among fragments through higher-level edges, making it more expressive than both standard node-level message passing and hierarchical methods that include fragment nodes but lack higher-level connectivity. Furthermore, our pretraining objective incorporates **embedding prediction**, enabling the model to capture richer semantic relationships across hierarchical levels.

---

> ### Author Response · Authors · 2025-11-26
>
> # Response to Weakness 3 & Q3
>
> We thank the reviewer for raising this point. As shown in **Appendix B (Figure 5 & 6)**, we have analyzed the distribution of extracted fragments. The analysis confirms that our heuristic rules align with chemical reality. The distribution of extracted rings is heavily dominated by 5- and 6-membered rings, which correspond to the most common stable chemical motifs (e.g., benzene, pyridine). Similarly, the "Path" fragments typically follow a distribution peaking at lengths of 2–4 atoms. These correspond to standard aliphatic linkers (e.g., ethyl, propyl chains) that determine molecular flexibility. These patterns confirm that our method does not produce arbitrary graph pieces but consistently recovers chemically meaningful, representative fragments aligned with known molecular structure regularities.
>
> To further explore the chemical representativeness of GraSPNet generated molecule representation, we conduct experiment following the same evaluation framework of MOLGRAPHEVAL[1]. We conduct probing task to predict fragment counts for the 23 common RDKiT fragments using the frozen GraSPNet generated molecule representation and report the accuracy for each fragment.
>
> **Table: Accuracy of GraSPNet in predicting counts of 23 common RDKit fragments (Part 1).**
>
> | RDKit Fragment | epoxide | lactam | morpholine | oxazole | tetrazole | NO | ether | furan | guanido | halogen | piperdine |
> |----------------|---------|--------|-------------|---------|-----------|----|--------|--------|----------|----------|------------|
> | **Accuracy (%)** | 99.5 | 99.95 | 99.45 | 99.10 | 98.85 | 99.60 | 91.75 | 99.60 | 99.55 | 94.35 | 95.70 |
>
> **Table: Accuracy of GraSPNet in predicting counts of 23 common RDKit fragments (Part 2).**
>
> | RDKit Fragment | thiazole | thiophene | urea | allylic oxid | amide | amidine | azo | benzene | imidazole | imide | piperzine | pyridine |
> |----------------|----------|-----------|-------|--------------|--------|----------|------|----------|------------|--------|------------|-----------|
> | **Accuracy (%)** | 99.05 | 99.70 | 99.75 | 95.45 | 94.65 | 99.65 | 99.80 | 92.25 | 98.75 | 99.40 | 97.20 | 97.15 |
>
> [1] Wang, Hanchen, et al. "Evaluating self-supervised learning for molecular graph embeddings." NeurIPS (2023)

---

### Official Review · Reviewer_duzi · 2025-11-06

**Soundness:** 3
**Presentation:** 2
**Contribution:** 2
**Rating:** 6
**Confidence:** 4

**Summary:**

This paper proposes GraSPNet, a novel self-supervised learning framework for molecular graphs that learns hierarchical representations by jointly predicting node and fragment-level embeddings. The goal is to improve molecular property prediction (for downstream tasks). This is done by capturing semantically rich pre-defined substructures (e.g., rings, paths, articulation points) during pretraining. Authors have shown that their proposed fragmentation strategy can effectively capture richer semantics, which will later be used for training their model.

**Strengths:**

1-	The paper proposed a novel approach to capture both node and fragment-level semantics. The proposed GraSPNet architecture introduces a dual-level semantic prediction mechanism, which is underexplored in graph self-supervised learning (GSSL).
2-	The proposed fragmentation strategy looks promising. Moreover, the WL-test example in Figure 2 clearly demonstrates how fragment-level abstraction helps distinguish structurally similar but semantically distinct molecules. This is a strong theoretical motivation.
3-	Authors have conducted an inclusive ablation study with respect to fragmentation. They have tested with and without fragmentation to demonstrate the effect of their proposed fragmentation strategy. Moreover, they have evaluated different fragmentation strategies to study the effectiveness of their proposed method compared to MGSSL, S-CGIB, and HiMOL methods.

**Weaknesses:**

1-	The baselines are outdated. Especially the graph contrastive learning methods. Here a list of GCL methods that have been published more recently and outperform current baselines:
GRACE: Zhu, Y., Xu, Y., Yu, F., Liu, Q., Wu, S., & Wang, L. (2020). Deep graph contrastive representation learning. arXiv preprint arXiv:2006.04131.
GCA: Zhu, Y., Xu, Y., Yu, F., Liu, Q., Wu, S., & Wang, L. (2021, April). Graph contrastive learning with adaptive augmentation. In Proceedings of the web conference 2021 (pp. 2069-2080).
GREET: Liu, Y., Zheng, Y., Zhang, D., Lee, V. C., & Pan, S. (2023, June). Beyond smoothing: Unsupervised graph representation learning with edge heterophily discriminating. In Proceedings of the AAAI conference on artificial intelligence (Vol. 37, No. 4, pp. 4516-4524).
EPAGCL: Xu, Y., Huang, S., Zhang, H., & Li, X. (2025, April). Why does dropping edges usually outperform adding edges in graph contrastive learning?. In Proceedings of the AAAI Conference on Artificial Intelligence (Vol. 39, No. 20, pp. 21824-21832).
2-	The authors have not included any analysis of training or inference cost as the graph size increases. Fragment graphs can become large and dense, but memory/runtime implications are not discussed.
3-	The paper needs an additional round of proofreading. There are several grammatical errors:
a.	Line 069: “at three semantic levels—node (atoms), fragment (e.g., functional groups)” -> “at three semantic levels: node (atoms), fragment (e.g., functional groups)”
b.	Line 182: GNNS -> GNNs
c.	Line 187: “can be more powerful than 2-WL test in distinguish graph isomorphic." -> “can be more powerful than the 2-WL test in distinguishing graph isomorphisms.”
d.	Ling 265: “to each nodes and fragments” -> “to each node and fragment”
e.	Line 811: “The code will be release upon acception.” -> “The code will be released upon acceptance.”

**Questions:**

a.	How do different fragmentation rules contribute to performance? It would be interesting to have an ablation study comparing different fragmentation schemes (e.g., rings-only, no articulation).

---

> ### Author Response · Authors · 2025-11-26
>
> We thank the reviewer for their constructive feedback and for recognizing the novelty of our dual-level semantic prediction and fragmentation strategy.
>
> # 1. Baselines (Weakness 1):
>
> The reviewer mentioned several competitive GCL methods. While these approaches (e.g., GRACE, GCA, GREET, EPAGCL) are indeed promising, they are primarily designed to **enhance node-level representation learning**, particularly by improving node discrimination and mitigating over-smoothing under heterophily. Consequently, they are mainly evaluated on node-centric benchmarks, such as **homophilic datasets** (Cora, CiteSeer, PubMed) and **heterophilic datasets** (Chameleon, Squirrel, Actor, Cornell, Texas, Wisconsin). These methods target challenges specific to node-level tasks like node classification and node clustering, which differ fundamentally from the **graph-level semantic modeling problem** addressed in our work.
>
> We agree that more promising graph contrastive learning baselines need to be added, so instead of the node-level baselines, we add new experiment of **graph-level baselines** including: GraphLOG[1], S2GAE[2], Mole-BERT[3], GraphMAE2[4]. The result are shown as follow:
>
> **Table: Comparison of ROC-AUC (%) on MoleculeNet benchmarks**
>
> | Method     | BBBP | Tox21 | ToxCast | Sider | MUV | HIV | Bace | Clintox |
> |------------|------|-------|---------|-------|-----|-----|------|---------|
> | GraphLOG   | 67.2 ± 1.3 | 76.0 ± 0.8 | 63.6 ± 0.7 | 59.8 ± 2.1 | 72.8 ± 1.8 | 72.5 ± 1.6 | 82.8 ± 0.9 | 76.9 ± 1.9 |
> | S2GAE      | 67.6 ± 2.0 | 69.6 ± 1.3 | 58.7 ± 0.8 | 55.4 ± 1.3 | 60.1 ± 2.4 | 68.0 ± 3.8 | 68.6 ± 2.1 | 59.6 ± 1.1 |
> | Mole-BERT  | 70.8 ± 0.5 | 76.6 ± 0.7 | 63.7 ± 0.5 | 59.2 ± 1.1 | 77.2 ± 1.1 | 76.5 ± 0.8 | 82.8 ± 1.4 | 77.2 ± 1.4 |
> | GraphMAE2  | 71.6 ± 1.6 | 75.9 ± 0.8 | 65.6 ± 0.7 | 59.6 ± 0.6 | **78.5 ± 1.1** | 76.15 ± 2.2 | 81.0 ± 1.4 | 78.8 ± 3.0 |
> | **GraSPNet (ours)** | **74.4 ± 1.5** | **77.3 ± 0.8** | **65.5 ± 0.5** | **62.5 ± 1.1** | **78.5 ± 1.3** | **78.0 ± 0.8** | **82.9 ± 3.1** | **84.1 ± 2.1** |
>
> [1] Xu, Minghao, et al. "Self-supervised graph-level representation learning with local and global structure." ICML. 2021.
>
> [2] Tan, Qiaoyu, et al. "S2gae: Self-supervised graph autoencoders are generalizable learners with graph masking." WSDM. 2023.
>
> [3] Xia, Jun, et al. "Mole-BERT: Rethinking Pre-training Graph Neural Networks for Molecules." ICLR 2023.
>
> [4] Hou, Zhenyu, et al. "Graphmae2: A decoding-enhanced masked self-supervised graph learner." Proceedings of the WWW. 2023.
>
> # 2. Analysis of Training/Inference Cost (Weakness 2)
> Thank you for pointing this out. We acknowledge that adding a fragment graph increases the topological complexity. However, we would like to highlight two key efficiency factors inherent to GraSPNet:
>
> -  **Coarsened Graph Size**: While we introduce a hierarchical structure, the fragment-level graph is generally much coarser than the atom-level graph (see Figure 5 in Appendix B), meaning the number of fragment nodes is significantly smaller than the number of atoms. Because we strictly generalize fragments into three types (rings, paths, articulation points), we avoid the large, sparse vocabularies common in other rule-based methods.
>
> -  **Deterministic Fragmentation**: Unlike methods that require expensive subgraph enumeration or frequent pattern mining (like MGSSL) during training, our fragmentation via RDKit is deterministic and computationally efficient.
> We have added Appendix G with a detailed runtime analysis. As shown in the table below, GraSPNet is highly efficient compared to advanced baselines:
>
>  **Table: Pretraining time comparison (in minutes)**
> | Model      | GraphMAE | Mole-BERT | S2GAE | GraphMAE2 | SimSGT | GraSPNet |
> |------------|----------|-----------|--------|------------|---------|-----------|
> | Pretrain Time | 527 | 2199 | 1763 | 1195 | 645 | 769 |
>
> GraSPNet contains a higher-level graph and message passing which makes it more expressive but slightly less efficient than GraphMAE and SimSGT which use only node-level graphs.
>
> # 3. Proofreading (Weakness 3):
>
> We have corrected all the specific grammatical errors listed (Lines 069, 182, 187, 265, 811) and conducted a thorough proofreading of the full manuscript.
>
> # 4. Fragmentation Ablation (Question a):
>
> We have added an ablation study in the Appendix comparing "Rings+Path" vs. our "Full Strategy" (Rings + Paths + Articulations). The results confirm that "Rings-Only" underperforms because it contains rings on higher-level graphs around the intersection node of “Paths”, which flattens the fragment representation during higher-level message passing. Our full strategy ensures comprehensive semantic coverage.
>
> **Table: Ablation study on articulation point extraction**
> | Type        | Clintox | BBBP | Sider | Bace | Mean |
> |-------------|---------|------|-------|------|-------|
> | w/o Arti    | 70.16   | 66.84 | 58.14 | **77.81** | 68.24 |
> | **Full**    | **70.35** | **67.30** | **59.61** | 76.86 | **68.53** |

---

### Meta-Review · Area_Chair_X4xv · 2026-01-06

**Summary:**

This submission proposes GraSPNet, a hierarchical self-supervised framework for molecular graphs that explicitly models atoms and “semantic fragments” (rings, paths, articulation points). The model learns by masking and predicting embeddings at both the node and fragment level, with message passing across atom–fragment and fragment–fragment interactions. The goal is to capture chemically meaningful structure beyond what standard node-level SSL typically picks up.

**Reviewer Concerns:**

Novelty/positioning: hierarchical and fragment/motif-based molecular pretraining has a lot of related work, and some reviewers felt this paper initially read like a solid integration rather than a clearly new idea.

Heuristic fragmentation: even without a learned vocabulary, the fragment definitions are still hand-designed, and it’s not fully obvious when this is the “right” granularity for different downstream tasks.

Efficiency and scaling: adding a fragment graph can increase complexity; reviewers wanted clearer runtime/memory discussion and comparisons to modern baselines.

Interpretability/chemical validity: reviewers asked for more concrete evidence that the extracted fragments and learned representations align with meaningful chemical motifs (beyond performance numbers).

**Reviewer Scores:**

Likely no change for the reviewers.

---

### Decision · Program_Chairs · 2026-01-26

Reject